

# Tropical convection regimes in climate models: evaluation with satellite observations

Andrea K. Steiner[1,2], Bettina C. Lackner[2], and Mark A. Ringer[3]

[1]Wegener Center for Climate and Global Change (WEGC) and Institute for Geophysics, Astrophysics, and
5 Meteorology/Institute of Physics, University of Graz, Graz, Austria
[2]Doctoral Programme Climate Change, University of Graz, Graz, Austria
[3]Met Office Hadley Centre, Exeter, U.K.

*Correspondence to*: Andrea K. Steiner (andi.steiner@uni-graz.at)

10 **Abstract.** High quality observations are powerful tools for the evaluation of climate models towards improvement and reduction of uncertainty. Particularly at low latitudes, the most uncertain aspect lies in the representation of moist convection and interaction with dynamics, where rising motion is tied to deep convection and sinking motion to dry regimes. Since humidity is closely coupled with temperature feedbacks in the tropical troposphere a proper representation of this region is essential. Here we demonstrate the evaluation of atmospheric climate models with satellite-based observations from Global 15 Positioning System (GPS) radio occultation (RO), which feature high vertical resolution and accuracy in the troposphere to lower stratosphere. We focus on the representation of the vertical atmospheric structure in tropical convection regimes, defined by high updraft velocity over warm surfaces, and investigate atmospheric temperature and humidity profiles. Results reveal that some models do not fully capture convection regions, particularly over land, and only partly represent high updraft or downdraft velocities. Models show large biases in tropical mean temperature of more than 4 K in the tropopause 20 region and the lower stratosphere. Reasonable agreement with observations is given in mean specific humidity in the lower to mid-troposphere. In moist convection regions, models tend to underestimate moisture by 10% to 30% over oceans whereas in dry downdraft regions they overestimate moisture by 100%. Our findings provide evidence that RO observations are a unique source of information, with a range of further atmospheric variables to be exploited, for the evaluation and advancement of next generation climate models.

## 25 1 Introduction

A key challenge in current representations of the climate system is the modelling of the atmospheric water cycle and its coupling with circulation. Despite continuous efforts in model development, the spread in model uncertainty remains large in current climate models, especially for regional projections of the hydrological cycle and potential changes to both the frequency and intensity of extreme events (Collins et al., 2013). Possible reasons for underlying model errors are that

30 processes and feedbacks are not well understood, not well represented or not well constrained by observations (Knutti and





Sedlacek, 2013). The Intergovernmental Panel on Climate Change (IPCC) finds differences in feedback processes as the main reason for the spread of model uncertainty stating in the latest world climate report: "*Water vapour, lapse rate and cloud feedbacks each involve moist atmospheric processes closely linked to clouds, and in combination, produce most of the simulated climate feedback and most of its inter-model spread*" (Boucher et al., 2013).

Differences in cloud feedbacks are considered to be the primary source of spread in both equilibrium and transient climate responses of the global mean surface temperature (the climate sensitivity) simulated by climate models (Dufresne and Bony, 2008). However, the contributions of water vapour and lapse rate are non-negligible, since their impact is reinforced by the mean model cloud feedback. Water vapour and temperature responses are closely coupled in the troposphere and models with a larger negative lapse rate feedback also have a larger positive water vapour feedback. The largest fractional changes

in specific humidity together with the largest feedback contribution occur in the tropical upper troposphere (e.g., Chung et al., 2014). Most models agree that the combined net feedback from water vapour and lapse rate changes is positive (Boucher et al., 2013) with the net effect resulting in the amplification of a warming, which is largest in the tropical middle and upper troposphere. Its proper representation in climate models is of central importance since it has more impact than other regions of the atmosphere.

Particularly at low latitudes, the most uncertain aspect lies in the representation of moist convection and its interaction with large-scale dynamics and improvements are necessary to reduce uncertainty in climate model projections (Stevens and Bony, 2013; Shepherd, 2014). This is one focus of the World Climate Research Programme (WCRP) and its grand challenge of clouds, circulation, and climate sensitivity. Bony et al. (2015) point in particular to enhance the understanding of cloud feedbacks and convective organization in climate and factors controlling it.

The distribution of humidity is determined by many factors, including the detrainment of vapour and condensed water from convective systems and the large-scale atmospheric circulation. The relatively dry regions of large scale descent play a major role in tropical long wave cooling, and changes in their area or changes in humidity could have a significant impact on water vapour feedback strength. Given the complexity of processes there is a need for modelling as well as observational studies (Randall et al., 2007). Observations are regarded essential for the development and evaluation of climate models. The

Observations for Model Intercomparison Projects (Obs4MIPs) (Teixeira et al., 2014) was launched for the sixth phase of the World Climate Research Programme's (WCRP) Coupled Model Intercomparison Project (CMIP6) (Meehl et al., 2014; Eyring et al., 2016) aiming for better use of observations (Gleckler et al., 2011; Teixeira et al., 2014).

Satellite observations from Global Positioning System (GPS) Radio Occultation (RO) are available since 2001 and provide atmospheric thermodynamic variables at high vertical resolution and accuracy (e.g., Scherllin-Pirscher et al., 2017) for

exploring the Earth's atmosphere, weather, and climate in the troposphere to the lower stratosphere (e.g., Anthes et al., 2011; Steiner et al., 2011). RO observations have significant impact in numerical weather prediction as they can be assimilated without bias correction, hence improving weather forecasts (e.g., Healy and Thépaut, 2006; Cardinali 2009) and anchoring atmospheric reanalyses (e.g., Poli et al., 2010; Simmons et al., 2014).





A range of studies on atmospheric variability and changes (e.g., Lackner et al., 2011; Scherllin-Pirscher et al., 2012; 2014; Randel and Wu, 2014) has shown that RO is highly useful for characterizing the tropopause (Randel et al., 2003; Schmidt et al., 2004; Borsche et al., 2007; Randel et al., 2014; Rieckh et al., 2014) and the thermodynamic structure during intense convection including cloud top detection (Biondi et al., 2012; 2015). Also the value of water vapor information from RO has

been demonstrated, e.g., for atmospheric dry layers (Rieckh et al., 2016) and for inferring water vapor feedback (Vergados et al., 2016). However, few studies exist on the evaluation of Global Climate Model (GCM) data with RO, so far comparisons of geopotential height (Ao et al., 2015) and of dry temperature climatologies (Kishore et al., 2016; Schmidt et al., 2016) have been performed.

Here we demonstrate the exploitation of GPS RO observations for the evaluation of tropical convection regions in GCM data

of the fifth Coupled Model Intercomparison Project (CMIP5). Rather than the usual approach of comparing observed and model climatologies at standard pressure levels (Ao et al., 2015; Kishore et al., 2016), we investigate the representation of temperature and humidity in the troposphere and lower stratosphere at highest available model resolution. We focus on the vertical structure of moist and dry regimes in the tropics and inspect their representation in models and observations.

## 2 Data

### 2.1 Radio occultation observations

GPS RO is an active limb sounding technique based on radio signals from Global Navigation System Satellites (GNSS) such as the GPS. On their way from the transmitter to a receiver on a Low Earth Orbit (LEO) satellite the microwave signals are refracted and retarded by the Earth's refractivity field. An occultation observation occurs if a GPS satellite sets behind (or rises from behind) the horizon and its signals are occulted by the Earth's limb from the viewpoint of the receiver. The

movement of the satellites enables a vertical scanning of the troposphere and stratosphere with high vertical resolution of about 100 m in the lower troposphere to about 1.5 km in the stratosphere (Kursinski et al., 1997; Gorbunov et al., 2004) but inherent along-ray horizontal averaging. The horizontal resolution across-ray is about 1.5 km and the along-ray resolution ranges from about 60 km in the lower troposphere to about 300 km in the stratosphere (Melbourne et al., 1994; Kursinski et al., 1997). Observations are made globally and under essentially all weather conditions.

The basic measurement is the signal phase as function of time, which is proportional to the optical path length between the transmitter and the receiver. The traceability to fundamental time standards with precise atomic clocks enables a long-term stable and consistent data record. The RO phase and amplitude measurements together with precise and accurate information on the satellites' orbits enable the retrieval of physical and thermodynamic variables including bending angle, refractivity, pressure, geopotential height, temperature, and specific humidity.

A main advantage is the independent provision of precise altitude information and pressure information (Scherllin-Pirscher et al., 2017), which allows the use of thermodynamic profiles at different vertical coordinates, e.g., temperature at mean-sea-level altitude, at geopotential height, or at pressure levels. Furthermore, tropopause parameters are provided with high



accuracy including tropopause temperature and tropopause height. The quality of RO measurements is best in the upper troposphere and lower stratosphere (UTLS). The observational uncertainty of individual temperature profiles is about 0.7 K in the tropopause region and slightly decreases toward the lower troposphere (Scherllin-Pirscher et al.; 2017). RO data from different missions are highly consistent and agree within 0.2 K between 4 km and 35 km for temperature (Scherllin-Pirscher

et al., 2011a). The data, from bending angle to temperature, can be merged without inter-calibration or homogenization, if the same processing system is used (Schreiner et al., 2007; Foelsche et al., 2011; Steiner et al., 2011; Angerer et al., 2017). Available RO data products include individual profiles as well as gridded climatologies (e.g., Ho et al., 2012; Steiner et al., 2013).

For this study we used individual profile data from the following RO missions: CHAllenging Minisatellite Payload

(CHAMP) (Wickert et al., 2001), Satélite de Aplicaciones Científicas (SAC-C) (Hajj et al., 2004), Gravity Recovery And Climate Experiment (GRACE-A) (Beyerle et al., 2005), and FORMOSAT-3/COSMIC (F3C) (Anthes et al., 2008). About 2000 globally distributed RO profiling measurements are available per day from these missions.

We used RO temperature and specific humidity profiles processed by the Wegener Center for Climate and Global Change (WEGC) with the Occultation Processing System (OPS) version 5.6 (Schwärz et al., 2016; Angerer et al., 2017), based on

excess phase and orbit data (versions 2009.2650 and 2010.2640) from the University Corporation for Atmospheric Research (UCAR). In the OPS retrieval, bending angle is initialized at high altitudes with background data from the European Centre for Medium-Range Weather Forecasts (ECMWF) short-range forecasts. Below 30 km, the retrieved bending angle profiles only contain observational information from RO. Bending angle is inverted to atmospheric refractivity (proportional to air density), which depends on the thermodynamic conditions of the dry and moist atmosphere, i.e., on pressure, temperature,

and water vapor pressure, given by the Smith-Weintraub formula (Smith and Weintraub, 1953; Kursinski et al., 1997) in Eq. (1):

$$N(z) = k_1 \frac{p(z)}{T(z)} + k_2 \frac{e(z)}{T^2(z)}, \qquad (1)$$

where the constants are $k_1$=77.6 K Pa$^{-1}$, $k_2$=3.73 × 10$^5$ K$^2$ Pa$^{-1}$, $p$ is pressure (in hPa), $T$ is temperature (in K), and $e$ is partial pressure of water vapor (in hPa). In dry air with very small to negligible moisture, as in the stratosphere, water vapor effects

are essentially negligible. The second term of Eq. (1) becomes zero and dry temperature profiles can be directly retrieved from the RO data.

In the troposphere, where moisture content is higher, temperature and specific humidity are retrieved based on optimal estimation of RO and ECMWF short-range forecast profiles. The background contributes relevant information for retrieved temperature profiles only in the lower to middle troposphere, when observed RO information is used to obtain humidity.

A detailed description of the OPS retrieval is given by Schwärz et al. (2016; Appendix A therein). Differences between retrievals of RO dry-air atmospheric profiles and moist-air tropospheric profiles are discussed by Scherllin-Pirscher et al. (2011b). Error estimates for RO profiles from bending angle to temperature are provided by Scherllin-Pirscher et al. (2011a; 2017), and for specific humidity by Kursinski et al. (1995; 1997), Steiner and Kirchengast (2005), and Kursinski and



Gebhardt (2014). RO humidity quality is best at low latitudes where the moisture concentrations in the lower and middle troposphere are highest. The accuracy of humidity profiles is estimated to about 0.3 g kg$^{-1}$ to 0.1 g kg$^{-1}$ in the lower to middle troposphere (Kursinski and Gebhardt, 2014) and to 0.03 g kg$^{-1}$ for climatological averages, which are also useful in the upper troposphere (Rieckh et al, 2016) where specific humidity is small. Excessive validation of RO atmospheric profiles

with independent observations, including radiosondes (e.g., Ladstädter et al., 2015) and satellite limb sounder data (e.g., Schwärz et al., 2016) has proved the high quality of RO variables.

## 2.2 Climate model data

For this study we used climate model data of the 5th phase of the Coupled Model Intercomparison Project (CMIP 5) (Taylor et al., 2012). Because our focus is the evaluation of atmospheric variables we chose the Atmospheric Model Intercomparison

Project (AMIP) experiments with atmosphere-only mode and prescribed sea surface temperature, which are available at higher vertical resolution. We selected those models available with a 6-hourly resolution in time and at model level resolution in the vertical, either at a hybrid sigma pressure grid or a hybrid height grid. Requested model variables included air temperature (ta), specific humidity (hus), and surface pressure (ps), the latter was needed for the computation of vertical pressure levels from the hybrid pressure grid. In addition, we selected the variables near surface air temperature (tas) and

lagrangian tendency of air pressure or vertical velocity (wap) at the 500 hPa level at daily resolution for classification purposes (see Sect. 3).

Eight different AMIP models were found available at 6-hourly time resolution. Out of these, finally five models had available all needed variables and all necessary information for comparison with RO observations either at altitude or pressure levels, i.e., for conversion of a hybrid sigma pressure grid to a pressure grid or for conversion of a hybrid height

grid to an altitude grid. The models used in this study are listed in Table 1, including model name, information on horizontal, vertical, and time resolution, modeling center and respective references.

We used the following models BCC-CSM1.1 of the Bejing Climate Center (BCC, China), CCSM4 of the National Center for Atmospheric Research (NCAR, USA), GFDL-CM3 of the National Oceanic and Atmospheric Administration Geophysical Fluid Dynamics Laboratory (NOAA/GFDL, USA), HadGEM2-A of the Met Office Hadley Center (MOHC, UK), and

NorESM1-M of the Norwegian Climate Center (NCC, Norway). The models' horizontal resolution ranges from near 1.25° x 0.95° to 2.5° x 2° in longitude and latitude. The number of vertical levels ranges from 26 to 48. Three models are available at 26 levels including BCC-CSM1.1, CCSM4, and NorESM1-M. HadGEM2-A is available at 38 levels and GFDL-CM3 at 48 levels of which 15 levels are above 10 hPa and not used in this study. For sensitivity tests we used in addition daily values of the HadGEM2-A model at a coarser vertical resolution with eight pressure levels only.

In addition, we used data of the European Centre for Medium-Range Weather Forecasts (ECMWF) Re-Analysis Interim (ERA-Interim) (Dee et al,. 2011) at 1.5° x 1.5° resolution in longitude and latitude with 6-hourly resolution, which is also reasonable for temporal collocation. Surface temperature and vertical velocity at 500 hPa from ERA-Interim were needed for



**Table 1.** Information on models: model name, horizontal, vertical, and time resolution, modeling center, and references.

| Model Name | Horizontal resolution [lon x lat] | Vertical resolution [no of levels] | Time resolution [hours] | Modeling Center | Reference |
|---|---|---|---|---|---|
| BCC-CSM1.1 | 1.875° x 1.865° (128 x 64) | 26 hybrid pressure | 6 hourly | Beijing Climate Center (BCC), China Meteorol. Administration | Wu et al. (2014) |
| CCSM4 | 1.25° x 0.95° (288 x 192) | 26 hybrid pressure | 6 hourly | National Center for Atmospheric Research (NCAR) | Gent et al. (2011) |
| GFDL-CM3 | 2.5° x 2° (144 x 90) | 48 hybrid pressure | 6 hourly | NOAA Geophysical Fluid Dynamics Lab (NOAA GFDL) | Griffies et al. (2011) |
| HadGEM2-A | 1.875° x 1.25° (192 x 145) | 38 hybrid height | 6 hourly | Met Office Hadley Centre (MOHC) | Martin et al. (2011) |
| NorESM1-M | 2.5° x 1.895° (144 x 96) | 26 hybrid pressure | 6 hourly | Norwegian Climate Centre (NCC) | Bentsen et al. (2013) |

the classification in updraft and downdraft regions (see Sect. 3). The reanalysis was taken as proxy for classifying the

observations because these variables are not provided by RO. The land-sea mask from ERA-Interim is used for classification

of all data sets over land and sea areas. ERA-Interim has a high horizontal resolution so that the spatial deviation when

collocating to RO tangent point locations is negligible.

## 3 Method

We investigate the representation of the vertical atmospheric structure in tropical moist and dry regimes in climate models

with respect to collocated RO observations based on temperature and specific humidity profiles. In our methodological

approach we therefore performed first a collocation of profiles and a systematic classification of different meteorological

regimes. The study region was limited to the tropics within 20°S to 20°N. In the vertical we focused on the troposphere to

the lower stratosphere region from about 850 hPa to 10 hPa where RO observations have the best quality. Temporal

constraints are given on the one hand by the RO observations, which are continuously available since May 2001, with an

increase in measurements since 2006 due to the F3C constellation, and on the other hand by the AMIP model data which are

provided only until December 2008. The inspected time period is thus 01/05/2001 to 31/12/2008.

### 3.1 Collocation of observations and model data

All available RO profiles for the defined time period and region were selected. The time and location information of each

individual RO profile, i.e., longitude and latitude of the tangent point and time of the RO event, was stored for collocation

with model data. The collocation procedure was based on a nearest neighbor approach. The collocated grid point in space




and time with respect to an RO event was calculated as the minimum distance between RO and model of the respective latitude, longitude, and time vector. For collocation in time, the different calendars of the models were taken into account as some models use 360 days per year (BCC-CSM1.1, CCSM4, NorESM1-M), 365 days per year (HadGEM2-A), or a standard calendar (GFDL-CM3). Collocated temperature and specific humidity profiles were extracted from RO and from the models

at the specific locations. Also ERA-Interim 6-hourly data were collocated and sampled at the RO event locations for classification of the observations.

In the vertical, the comparisons of model versus observation were performed at pressure levels and at altitude levels. In order to compare with models that were available at hybrid pressure levels (see Table 1), we converted those to pressure levels. We used 24 levels from 1000 hPa to 10 hPa including the 17 pressure levels as defined by the World Meteorological

Organization plus additional levels in the tropopause region. The HadGEM2-A model data were available at 38 hybrid height levels and the comparison was performed at a mean-sea-level altitude grid from 100 m to 33 km. The latter enabled us to perform a comparison at high vertical resolution. In addition we compared to HadGEM2-A at a very coarse vertical resolution with only 8 pressure levels for testing the sensitivity to the vertical resolution.

### 3.3 Classification of dynamical regimes

The classification of dynamical regimes is based on the fact that regions of rising motion (upper level divergence) are closely tied to regions of deep convection, whereas regions of sinking motion (convergence) represent mean clear sky conditions (Lau et al., 1997). In regions of large-scale ascending motions optically thick convective clouds occur over the warmest ocean waters, e.g., the western Pacific warm pool or the eastern Indian Ocean, having strong effects in both the short-wave and long-wave parts of the spectrum, with large reductions of outgoing long-wave radiation (OLR) and large increases in the

shortwave reflection to space. Areas of subsidence are associated with both clear sky conditions and areas of low-level boundary layer cloud, which form over cooler sea surface temperatures (SST), e.g., the equatorial Pacific cold tongue. The low-level cloud has a strong short-wave cooling effect but only a weak long-wave effect (e.g., Ringer and Allan, 2004).

According to Bony et al. (1997) pressure vertical velocity at 500 hPa ($\omega_{500}$) is a good proxy for the large-scale vertical motion (ascending/descending air) associated with the large-scale tropical circulation. In the tropics, the $\omega_{500}$ is proportional

to the 200 hPa divergence and to the 850 hPa convergence. It corresponds to the change in pressure $p$ with time $t$, $\omega = dp/dt$, with rising motions corresponding to negative values of $\omega_{500}$ whereas sinking motions correspond to positive values of $\omega_{500}$. The occurrence of large-scale rising motion is furthermore strongly coupled with SST of larger than about 26°C to 27°C. For SST less than about 26°C to 27°C, one finds sinking motions over the tropical oceans with low-level cloudiness over this region (Bony et al., 1997; Ringer and Allan, 2004). According to Ringer and Allan (2004), higher SSTs of larger than 26.5°C

are associated with a wide range of vertical velocities from strongest ascent to strongest descent, whereas lower SSTs of less than 26.5°C are primarily associated with strong descent or only very weak vertical motion.

In this study we therefore classify dynamical regimes of large-scale atmospheric motion in terms of vertical motion $\omega_{500}$ and near surface air temperature ($T_{2m}$). The regimes range from strong ascent defined by $\omega_{500} < -40$ hPa d$^{-1}$ to strong descent




defined by $\omega_{500} > 40$ hPa d$^{-1}$ and from coolest waters to warmest waters with the boundary between the two regimes defined at $T_{2m}$ of 26°C. The $\omega_{500}$ classes range from –160 hPa d$^{-1}$ to +160 hPa d$^{-1}$ with class intervals of 20 hPa d$^{-1}$. The $T_{2m}$ classes range from 17.5°C to 32.5°C with class intervals of 1°C. In addition, a distinction between regions over land and over oceans is made. Ocean or land was assigned first, then data were classified with respect to $\omega_{500}$ and $T_{2m}$ (after Ringer and Allan,

5    2004).

According to the defined classifications, we sorted and sampled RO profiles and model profiles over the period 05/2001 to 12/2008 on a daily basis. For the RO data, the respective classes were defined using the ERA-Interim dynamical fields ($\omega_{500}$, $T_{2m}$; not available from RO) as proxy. Information on daily mean $\omega_{500}$ values and $T_{2m}$ was extracted from ERA-Interim at the RO locations. RO temperature and humidity profiles were then sampled into the respective classes. This provides

information on the distribution of updraft and downdraft regimes in the observations. For the model data, classes were assigned using each model's own dynamical fields ($\omega_{500}$, $T_{2m}$). Information on daily mean $\omega_{500}$ and $T_{2m}$ from the models was extracted at (collocated) RO locations. Model temperature and humidity profiles were then sampled into the respective classes. After sampling temperature and humidity profiles in updraft and downdraft regimes, a detailed evaluation of the occurrence distributions in models and observations was performed and differences in models with respect to the

observations were investigated.

Figure 1 shows the representation of tropical convection regions in observations (ERA-Interim) and in two exemplary models, HadGEM2-A and CCSM4 (at the locations of RO observations) in form of Hovmöller diagrams. Convection regions are denoted by updraft (negative $\omega_{500}$ in blue) and found in ERA-Interim (Fig. 1a) from 30°W to 90°W over Amazonia, from 0° to 30°E over equatorial Africa, and largest from 60°E to 180°E over Indonesia from the eastern Indian ocean to the

western Pacific, corresponding to regions of low OLR. Non-convection regions denoted by sinking motion (positive $\omega_{500}$ in red) are found over the eastern Pacific and Atlantic regions, corresponding to regions of high OLR. Comparing the representation of the respective regions, we find that some models do not fully capture convection regions, e.g., HadGEM2-A shows a gap between 90°E to 130°E over Indonesia (Fig. 1b) and GFDL_CM3 over Africa (not shown), while others such as CCSM4 (Fig. 1c) are in better agreement with ERA-Interim. Such differences may be related to the representation of

tropical variability phenomena, e.g., the simulation of the 30 day to 60 day (Madden-Julian) equatorial oscillation.



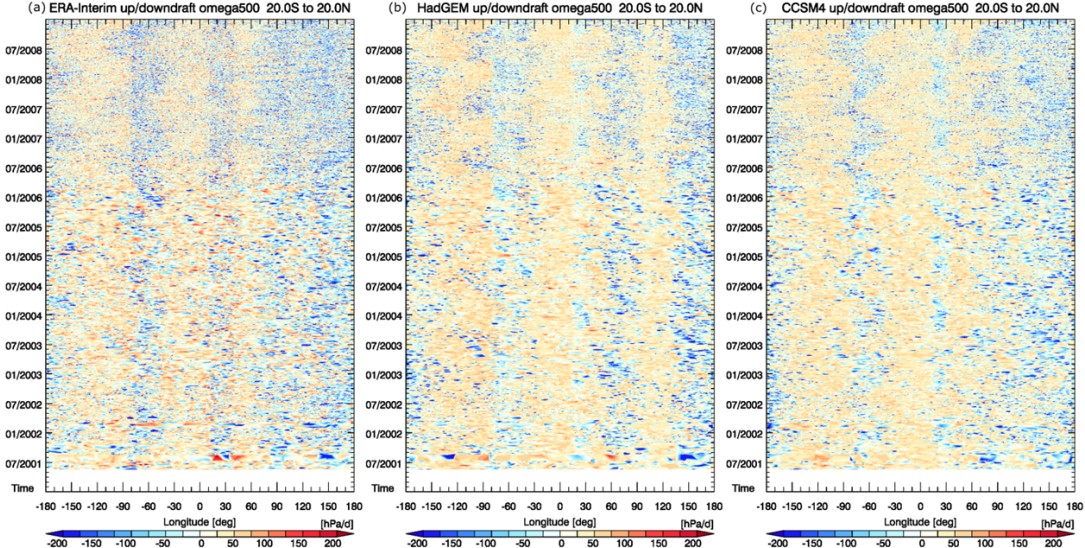

**Figure 1.** Tropical convection regions represented as function of time versus longitude in ERA-Interim as proxy for the observations (a), the HadGEM2-A model (b), and the CCSM4 model (c), shown for co-locations with RO observations getting denser in 2006. Updraft (blue) denotes convection regions and downdraft (red) denotes non-convection regions.

An overview on the distribution of $\omega_{500}$ and $T_{2m}$ in ERA-Interim and in the different models is presented in Fig. 2. Most models show a narrower distribution in $\omega_{500}$ than ERA-Interim, underestimating velocities near ±50 hPa s$^{-1}$ (except for HadGEM2-A) and overestimating small velocities near ±20 hPa s$^{-1}$ (Fig. 2a). This feature is found to be more distinct over the ocean (Fig. 2c) but is also seen over land (Fig. 2e). An exception is the BCC_CSM1.1 model which shows a similar form of the distribution as ERA-Interim but is shifted towards higher velocities. For the temperature distributions quite good agreement is found in models with respect to ERA-Interim (Fig. 2b), especially over sea (Fig. 2d). This most likely results from the prescription of observed sea surface temperature (SST) as boundary condition in the AMIP simulations. A slight shift is seen in the GFDL_CM3 model and the BCC_CSM1.1 model towards higher temperatures, particularly over land (Fig. 2f). There, the temperature distribution in GFDL_CM3 is shifted by about 1°C to 2°C and BCC_CSM1.1 shows a smoother and broader temperature distribution than other models, thereby underestimating mean temperatures and overestimating higher temperatures above 26°C.



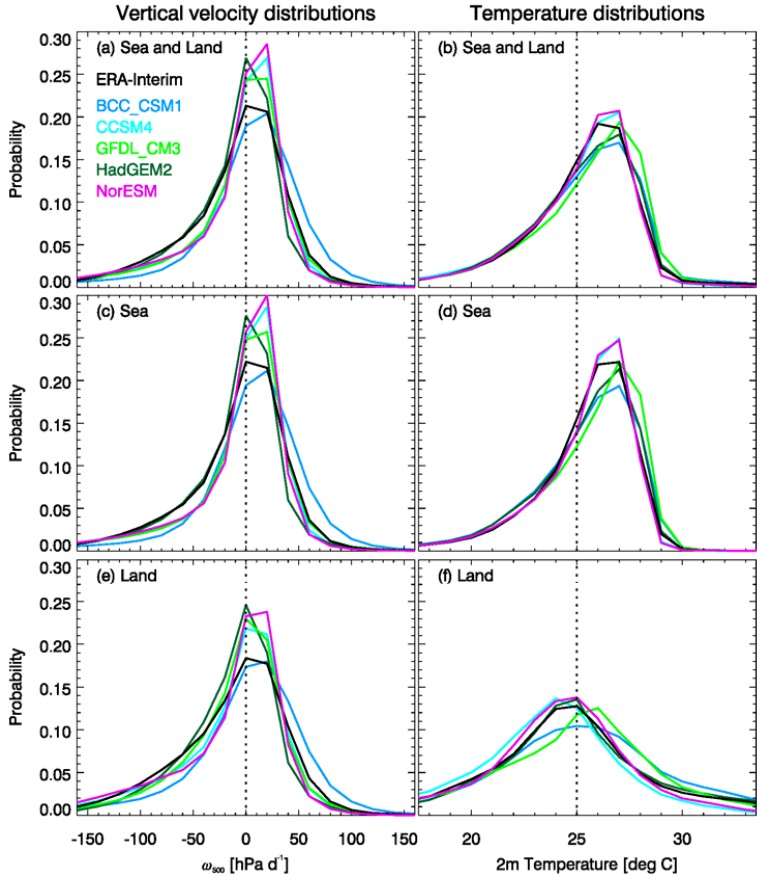

**Figure 2.** Distribution of vertical velocity $\omega_{500}$ (left) and near surface temperature $T_{2m}$ (right) in the tropics over sea and land (a, b), over sea only (c, d), and over land (e, f) shown for ERA-Interim as proxy for the observations (black) and for five climate models (colored).

### 4 Results and discussion

5    The distribution of atmospheric profiles from RO observations and from five different models into vertical velocity classes and temperature classes is detailed in Fig. 3. More than 400000 collocated profiles are found in total over sea and land (left panels) of which about 80% are distributed over sea (middle panels) and about 20% over land (right panels). RO observations show a slightly lower number of occultations over land with 19.6% compared to the models ranging from 20.7% to 22.1%.



Different distribution patterns are found for models compared to the observations over sea as well as over land. While RO observations are mainly clustered (more than 3000 profiles per class) in vertical velocity from –60 hPa d$^{-1}$ to 80 hPa d$^{-1}$ and from 23°C to 28°C over sea, the models show a more narrow distribution in vertical velocity from –20 hPa d$^{-1}$ to 60 hPa d$^{-1}$ but a broader distribution in temperature from 21°C to 29°C. The models are missing profiles in updraft classes of

< –20 hPa d$^{-1}$ between 24°C to 26°C. Also classes above 80 hPa d$^{-1}$ in vertical velocity are hardly occupied by model profiles. For BCC_CSM1.1 the profile distribution over sea is a bit more similar to the observations than that of the other models, although classes of large updraft and high surface temperature (<–40 hPa d$^{-1}$ and > 25°C) are hardly occupied. In general, the results are consistent with the findings of Ringer and Allan (2004) that lower temperatures (SST < 26°C) are associated with only weak vertical motion or strong descent.

Over land the distributions are quite different. RO profiles are clustered in classes between ±40 hPa d$^{-1}$ and 22°C to 27°C whereas model profiles are broader distributed in temperature classes from 21°C to 29°C with the maximum occurrence shifted to higher temperatures of 25°C to 26°C. Note that using a different reanalysis data set for vertical velocity and near surface temperature might give a slightly different distribution of the observed RO profiles.

### 4.1 Climatological differences between models and observations in the tropics

Having sampled atmospheric profiles into the respective updraft and downdraft classes we inspected the differences between temperature profiles and humidity profiles in models with respect to the RO observations. We computed the mean temperature profile and mean humidity profile of each class for models and observations, respectively, and then the difference profiles of model minus RO. In a first test we investigated different model resolutions.

We compared the HadGEM2-A model at a coarse vertical resolution using four pressure levels only (250 hPa, 100 hPa, 50

hPa, 10 hPa) (Fig. 4a). The overall mean difference in (dry) temperature (see Sect. 2.1) between HadGEM2-A and RO shows a warm bias of the model of about 3.5 K at 100 hPa and of 2.2 K at 50 hPa in the stratosphere, whereas a cold bias of –1 K is seen at 250 hPa in the troposphere. These biases are consistent with those shown by Hardiman et al. (2015). In particular, the UTLS biases are long-standing issues in the Hadley Centre model, including HadGEM2 shown here, and have been considerably reduced in the most recent version (HadGEM3), which will be submitted for CMIP6. Noticeable is the large

standard deviation in the tropopause region. We further compared to HadGEM2-A at a (four times) higher vertical resolution using 16 altitude levels (Fig. 4b) and found very much the same picture regarding the bias but a much smaller variance.

Comparing (physical) temperatures (see Sect. 2.1) shows consistent results (Fig. 4c), which is expected as physical temperatures are identical to dry temperature in a dry atmosphere, i.e., in the upper troposphere to lower stratosphere (see Sect. 2.1). Mean temperature differences show that HadGEM2-A is about –1 K colder than RO in the troposphere. It is about

4 K warmer in the tropopause region and about 2 K warmer in the lower stratosphere region. Inspecting absolute temperature





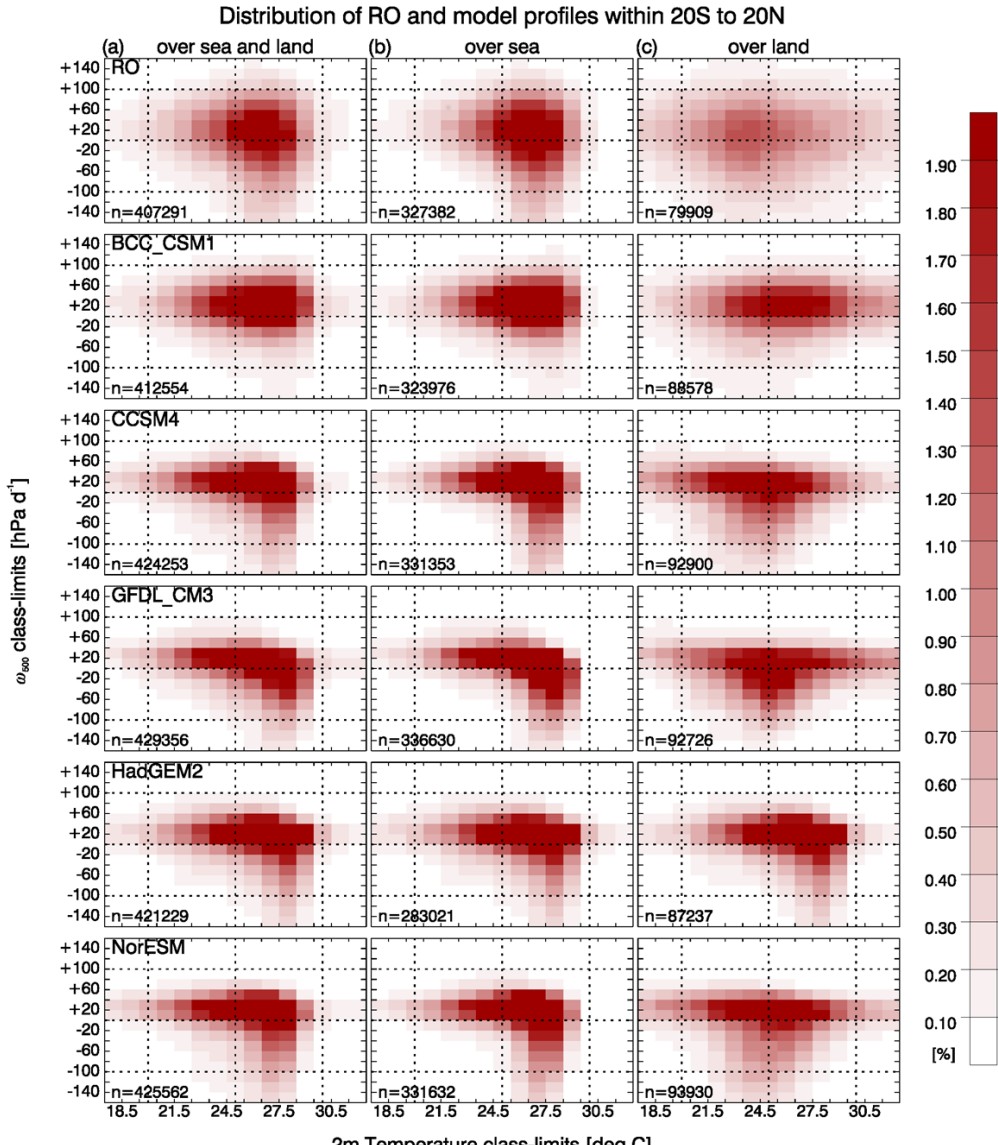

**Figure 3.** Data distribution in vertical velocity classes $\omega_{500}$ and near surface temperature classes $T_{2m}$ over sea and land (a), over sea (b), and over land (c) shown for RO observations and five climate models, BCC-CSM1.1, CCSM4, GFDL-CM3, HadGEM2-A, NorESM1-M (top to bottom). For better perception, the gridded lines mark temperatures of (20, 25, 30) °C and $\omega_{500}$ at (−100, 0, 100) hPa d$^{-1}$.





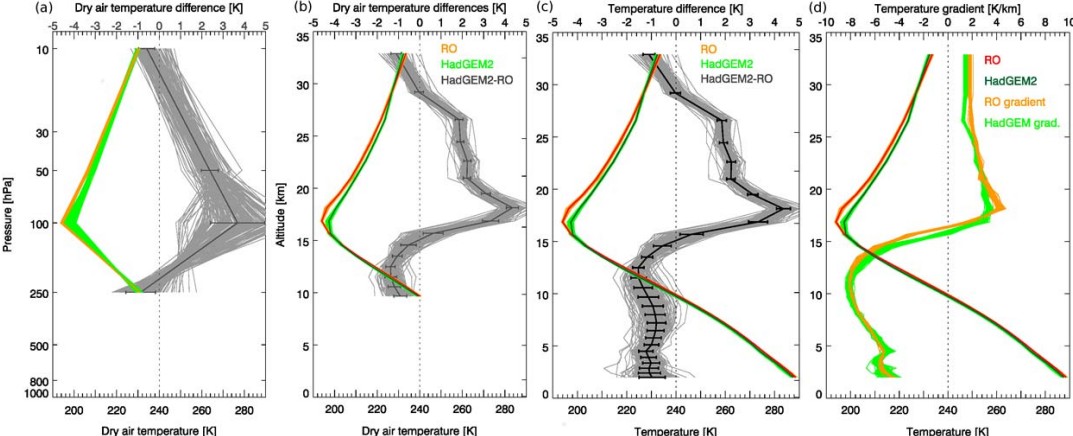

**Figure 4.** Comparison of HadGEM2-A model data to RO observations for all classes ($\omega_{500}$ and $T_{2m}$) in the tropics in terms of collocated temperature profiles and temperature difference profiles of HadGEM2-A minus RO (lower axes depict absolute values, upper axes differences to RO). Shown are dry temperature (see Sect. 2.1) at coarse vertical resolution for four pressure levels (a), dry temperature at high vertical resolution for altitude levels (b), as well as physical temperature at high vertical resolution (c) and corresponding temperature gradients (d).

profiles and the respective vertical temperature gradients (Fig. 4d) reveals that the differences stem from a sharper representation of the tropopause in the observations and from larger temperature gradients in the upper troposphere and in the lower stratosphere (Fig. 4d), which the model does not capture. The results furthermore indicate that a higher resolution does not help to reduce the large model bias in the tropopause region but to reduce variance.

In the following evaluations we use the highest available vertical model resolution and compare at two different vertical grids (see Sect. 2.2). We compare the HadGEM2-A model at altitude and the other models at pressure levels to RO. Figure 5 presents absolute temperature profiles and difference profiles between model and observations. The mean absolute temperature over all classes provides a representation of the climatological mean temperature in the tropics. It decreases from 290 K near 850 hPa to about 195 K in the tropopause and increases to 230 K near 10 hPa. In the troposphere, performance is best for NorESM-M and also for CCSM4 and GFDL_CM3, whereas BCC_CMS1.1 shows a large negative bias of up to –3 K near 250 hPa and HadGEM2-A a constant bias of –1 K with respect to RO. Large temperature differences are revealed in the tropopause region for all inspected models, ranging from –4.5 K in CCSM4 to 4.4 K in HadGEM2-A. In comparison the maximum difference between RO and ERA-Interim is about –1 K in the tropopause region and less than 0.5 K in the troposphere and lower stratosphere.





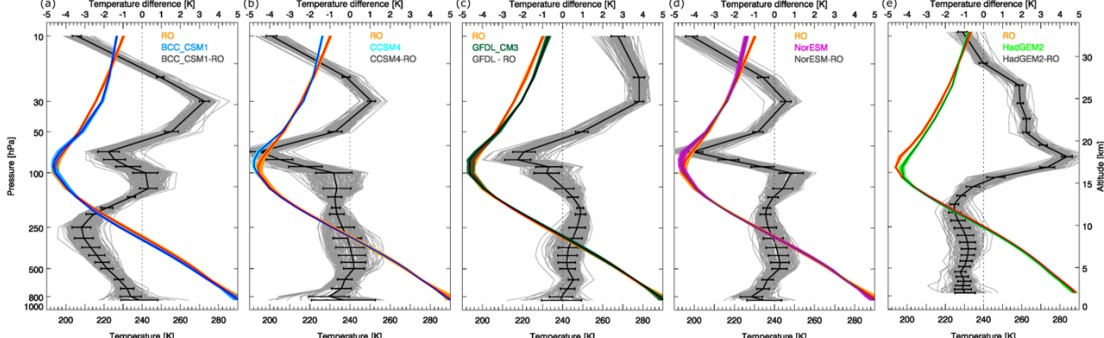

**Figure 5.** Temperature difference of model data minus RO observations (upper axes) and collocated temperature profiles for all classes ($\omega_{500}$ and $T_{2m}$) in the tropics (lower axes), shown for BCC-CSM1.1 (a), CCSM4 (b), GFDL-CM3 (c), and NorESM1-M (d) at pressure levels, and for HadGEM2-A (e) at corresponding altitude levels.

In the lower stratosphere, most models are too warm, with biases of 2 K to 4 K. This is again consistent with Hardiman et al (2015) and also with Kim et al. (2013), both of whom evaluated climate models against reanalysis temperatures. The results on differences in tropical temperature climatology agree with findings of Kishore et al. (2016, Fig. 3a therein), who also used GPS RO data. However, our analysis indicates that the tropopause is not well captured in models. The large differences

possibly result from the representation of the tropopause at specific model levels, in BCC_CSM1.1 at 70 hPa, in CCSM4 at 80 hPa, in GFDL_CM3 at 100 hPa and in NorESM1-M at 70 hPa. Furthermore, temperature gradients in models differ from those in the RO observations in the upper troposphere and the lower stratosphere. Note that Hardiman et al. (2015) have suggested that model biases in the height of the tropopause might be related to the dipole in the temperature bias between the upper troposphere and lower stratosphere. Clearly this is an area where the RO measurements should potentially be of great

use, especially as the vertical resolution of models increases and the ability to resolve the UTLS temperature gradients improves.

For humidity profiles (Fig. 6) quite good agreement is given in tropical mean specific humidity. BCC-CSM1.1 agrees well with RO. CCSM4, GFDL_CM3, and NorESM-M show a positive humidity bias of 10% to 20% and HadGEM2-A a negative humidity bias of 10% to 20% in the lower to middle troposphere. Differences get larger in the upper troposphere where the

amount of humidity is very low. RO specific humidity profiles show larger variance across the classes while the models do not show such a large spread in humidity.




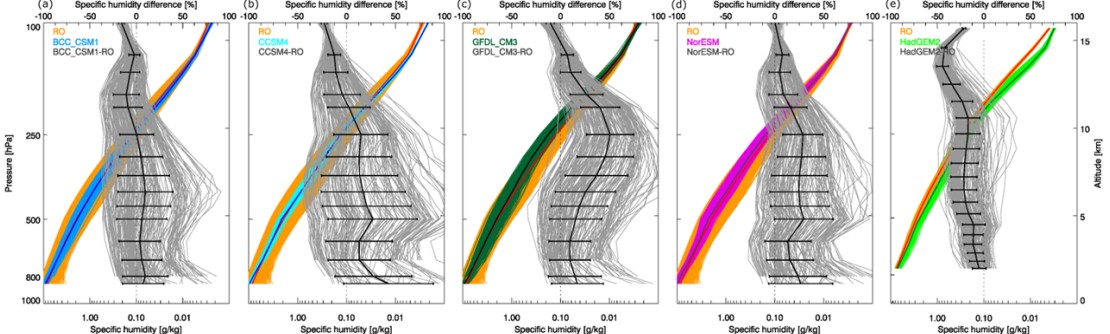

**Figure 6.** Specific humidity difference of model data minus RO observations and collocated specific humidity profiles for all classes ($\omega_{500}$ and $T_{2m}$) in the tropics, shown for BCC-CSM1.1 (a), CCSM4 (b), GFDL-CM3 (c), and NorESM1-M (d) at pressure levels, and for HadGEM2-A (e) at corresponding altitude levels. Note the different y-axis compared to Fig. 5 for temperature.

## 4.2 Humidity differences in convection regions and non-convection regions

A separate evaluation of convection regions and non-convection region was performed in order to gain a better understanding of the differences seen in humidity in these selected regions. Convection regions with strong updraft were defined by $\omega_{500} < -40$ hPa d$^{-1}$ and $T_{2m} > 26°C$ corresponding to classes in the lower right part of Fig. 3a. Non-convection

regions with strong downdraft were defined by $\omega_{500} > 40$ hPa d$^{-1}$ and $T_{2m} < 26$ C corresponding to classes in the upper left part of Fig. 3a. All classes falling into the specified regions were selected.

Specific humidity is found about two to three times larger in convection regions than in non-convection regions, particularly over sea. Specific humidity in the specified convection region ranges from about 17 (14) g/kg at 1000 hPa, 10 (8.8) g/kg at 800 hPa, 3 (2.5) g/kg at 500 hPa, to 0.07 (0.05) g/kg at 200 hPa over the ocean (over land).

Figure 7 shows the respective classified humidity profiles in convection regions (left panels) and in non-convection regions (right panels) separately over sea (left subpanels) and over land (right subpanels) for each model (top to bottom). For specific humidity in convection regions (Fig. 7, left panels) quite reasonable agreement between models and observations is found and the variance is small. Over land, the agreement is best with mean differences of less than 10%, except for GFDL_CM3. In convection regions over the ocean, most models tend to underestimate moisture by 10% to 30%. HadGEM2-A shows the

best performance in representation of specific humidity, while GFDL_CM3 shows a large bias near 250 hPa.



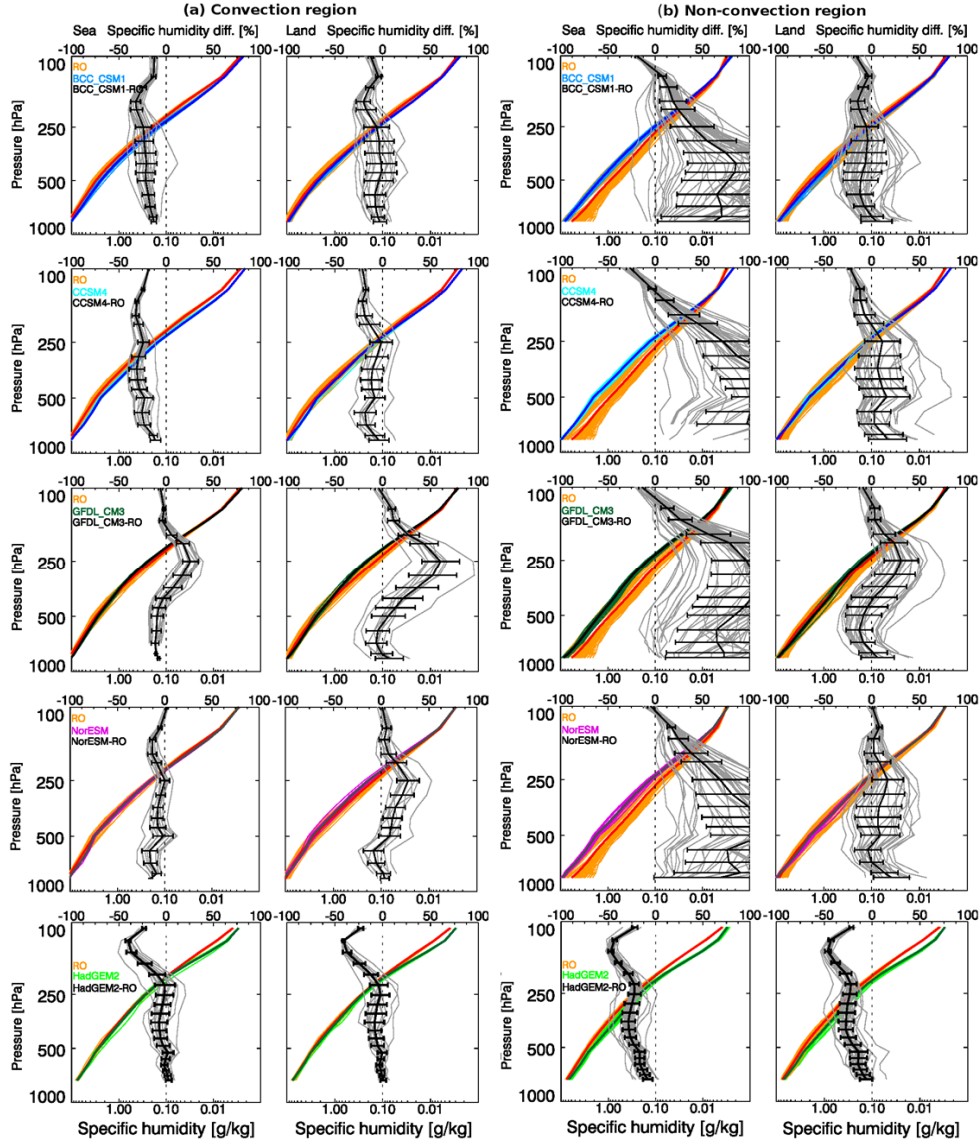

**Figure 7.** Specific humidity difference of model data minus RO observations and collocated specific humidity profiles for BCC-CSM1.1, CCSM4, GFDL-CM3, NorESM1-M, and HadGEM2 (top to bottom), shown for convection regions (a) over sea (left) and over land (right) and for non-convection regions (b) over sea (left) and over land (right).





In non-convection regions (Fig. 7, right panels) the amount of humidity over sea is much smaller and thus the spread in differences is larger. Best agreement with observations is given over land for three models, BCC-CSM1.1, CCSM4, and NorESM1-M. HadGEM2-A is about 25% too dry, GFDL is too moist near 250 hPa. Largest differences are found in non-
convection regions over the ocean where all models, except HadGEM2-A, are found too moist and overestimate specific humidity by 100%. This result is consistent with findings by Kursinski and Gebhard (2014) showing that models have shortcomings in representing dry extremes.

## 5 Summary, conclusions, and outlook

Tropical convection regimes in CMIP5 atmospheric climate models were evaluated with satellite-based observations from
GPS radio occultation (RO), which feature high vertical resolution and accuracy in the upper troposphere to lower stratosphere. Focusing on moist and dry regimes in the tropics we investigated the representation of these regimes in models with respect to the observations. We inspected the vertical atmospheric structure based on temperature and specific humidity in the troposphere and lower stratosphere using five AMIP models available with high resolution in the vertical (model levels) and in time (6-hourly), including BCC-CSM1.1 (BCC, China), CCSM4 (NCAR, USA), GFDL-CM3 (NOAA/GFDL,
USA), HadGEM2-A (MOHC, UK), and NorESM1-M (NCC, Norway). Comparison with GPS RO was carried out for different vertical resolutions and different vertical grids, at pressure levels as well as altitude levels.

We performed a systematic classification of moist updraft and dry downdraft regimes based on pressure vertical velocity at 500 hPa ($\omega_{500}$) and near surface air temperature ($T_{2m}$), since regions of rising motion over warm surfaces are tied to deep convection and regions of sinking motion over cooler surfaces are tied to dry regimes. ERA-Interim was used as proxy for
defining classes in the observations, which range from regimes of strong ascent ($\omega_{500} < -40$ hPa d$^{-1}$ and $T_{2m} > 26°C$) to regimes of strong descent ($\omega_{500} > 40$ hPa d$^{-1}$ and $T_{2m} < 26°C$). According to the defined classifications, collocated model profiles and RO profiles were sorted and sampled over the period 05/2001 to 12/2008 on a daily basis. Computation of differences in collocated profiles made deviations between models and observations apparent.

Regarding the representation of updraft and downdraft regions over longitude, we found that not all models fully capture
updraft, e.g., HadGEM2-A shows a gap over Indonesia and GFDL_CM3 over Africa, while other models are in better agreement with ERA-Interim. Most models underestimate higher vertical velocities near ±50 hPa s$^{-1}$ (except for HadGEM2-A) but overestimate low vertical velocities near ±20 hPa s$^{-1}$. Surface temperature distributions are in quite good agreement with ERA-Interim, which is most likely due to the prescription of observed sea surface temperature as boundary condition in the AMIP models. However, particularly over land, some models (BCC_CSM1.1, GFDL_CM3) show a shift to higher
surface temperatures.

Over sea, atmospheric profiles of RO observations are clustered from –60 hPa d$^{-1}$ to 80 hPa d$^{-1}$ in vertical velocity and from 23°C to 28°C in surface temperature. Model profiles are clustered narrower from –20 hPa d$^{-1}$ to 60 hPa d$^{-1}$ but over a broader



temperature range from 21°C to 29°C. Updraft classes with higher velocities (< –20 hPa d$^{-1}$) over lower surface temperatures and downdraft classes with high vertical velocities (> 80 hPa d$^{-1}$) are hardly occupied in models in contrast to the observations. Over land, the distribution of RO and model profiles is quite different. RO profiles are clustered in classes between ±40 hPa d$^{-1}$ and 22°C to 27°C, whereas model profiles are broader distributed over surface temperature and shifted

to higher temperatures.

Mean temperature differences between models and observations in the tropics show quite good agreement of about 1 K in the troposphere (with exceptions) but large model biases are revealed in the tropopause region reaching more than 4 K. Also in the lower stratosphere, most models exhibit a warm bias of 2 K to 4 K. These results show that the tropopause region is not well captured in models, which is due to the little variability in tropopause height in models defined on standard pressure

levels and too small temperature gradients in the tropopause region and the lower stratosphere (compared to sharp tropopause structures and steeper temperature gradients in RO). For humidity in the tropics we found reasonable agreement with differences of 10% to 20% in mean specific humidity in the lower to middle troposphere.

Inspecting convection regions, where the amount of humidity is large, we found reasonable agreement between models and observations in the lower to middle troposphere over land. Over the oceans most models underestimate moisture. Specific

humidity differences of about only 10% are found for HadGEM2 and NorESM, whereas BCC-CSM1.1 and CCSM4 are about 20% to 30% too dry over sea during convection and GFDL_CM3 shows a large moist bias above 500 hPa.

In non-convection regions, the amount of humidity over sea is much smaller and the spread in differences is found larger. Over land, most models agree well with observations within about 10%, except HadGEM2-A is about 25% too dry. Largest differences are found in non-convection regions over sea where almost all models are too wet and overestimate observed

moisture by 100%.

Our results provide evidence of the value of RO observations for model evaluation in the troposphere to lower stratosphere as we demonstrated for temperature and specific humidity. RO delivers vertically high resolved temperature profiles for inspecting the thermal structure in models which might be particularly useful for improving the models' representation of the tropopause region. Humidity information from RO in the tropical troposphere is valuable for identifying biases in models as

shown here for the case of convection and non-convection regimes. A number of dynamical, microphysical, and radiative processes influence the simulation of both the tropical tropopause temperature and the lower-stratospheric water vapor in climate models (Hardiman et al., 2015) and the RO observations provide a valuable source of information to help evaluate and improve their representation.

A range of further atmospheric variables is available from RO, such as e.g., bending angle, refractivity, density, tropopause

temperature and tropopause altitude. The advantage of RO lies in the high vertical resolution, high accuracy and precision, global availability, and virtually independent information on altitude and thermodynamic atmospheric variables, which can be used at different vertical coordinate systems including mean sea level altitude, geopotential height, pressure, or potential temperature (Scherllin-Pirscher etal., 2017).



This enables evaluating models not only at standard pressure levels as currently foreseen in Observations for Model Intercomparison Projects (Obs4MIPs) but at best possible model resolution, thus exploiting RO observations for specified comparison tasks as stated by Notz (2015) "*climate models can only meaningfully be evaluated relative to a specific purpose*". A possible way could be to perform model evaluation in observational space, e.g, bending angle space, by

converting model information with a forward model operator and thereby making use of the full information content of RO measurements independently of retrieved variables. Work is currently underway to include RO forward models in the GCM satellite simulator COSP (Bodas-Salcedo et al., 2011).

The RO community (www.irowg.org) is making RO observations from several processing centers available for Obs4MIPs. This is a common undertaking within the project *Radio Occultation based gridded CLIMate data sets (RO-CLIM)* in the

World Meteorological Organization's program on Sustained, Coordinated Processing of Environmental Satellite data for Climate Monitoring (SCOPE-CM) (http://www.scope-cm.org/projects/scm-08/).

In the near future several new RO missions are planned like the COSMIC-2 constellation as well as the use of signals from several Global Navigation Satellite Systems (GNSS), e.g., the Chinese BeiDou system, the European Galileo and Russian Glonass systems. Thus the number of occultation observations is expected to largely increase in the next years. This will

enhance the importance of RO as a unique observational data set and as a new source of information for evaluation, development and testing of the next generation of climate models.

## 6 Data availability

The climate model data used in this study are publicly available from the CMIP 5 database of the Program for Climate Model Diagnosis and Intercomparison (PCMDI) established at the Lawrence Livermore National Laboratory (LLNL) and

can be accessed via <http://cmip-pcmdi.llnl.gov/cmip5/>. ERA-Interim data used in this study are publicly available from ECMWF (Reading, United Kingdom) and can be accessed via <https://www. ecmwf.int>. The WEGC OPSv5.6 RO data are available on request from A. K. Steiner and will be made publicly available soon.

*Author contributions*. A.K.S. performed the main data analyses, produced the graphics, and wrote the manuscript. B.C.L.

wrote basic programming code, performed initial analyses, and contributed to the manuscript. M.A.R. provided guidance on study design and model aspects, and contributed to finalizing the manuscript.

*Competing interests*. The authors declare that they have no conflicts of interest

*Acknowledgements*. We are grateful to G. Kirchengast (WEGC, AT) for initial ideas for this study and a range of further inputs and valuable discussions in course of the work. B. Scherllin-Pirscher (WEGC, AT) is thanked for many fruitful




discussions and helpful comments. M. W. Jury (WEGC, AT) is thanked for help on model data collection. We are grateful to the UCAR/CDAAC (Boulder, CO, USA) for the provision of level 1a RO data and ECMWF (Reading, UK) for access to its re-analysis, analysis, and short-term forecast data. We thank the WEGC processing team members, especially M. Schwärz (WEGC, AT), for OPSv5.6 RO data. We acknowledge the World Climate Research Programme's Working Group on

Coupled Modelling, which is responsible for CMIP, and we thank the climate modeling groups (listed in Table 1 of this paper) for producing and making available their model output. For CMIP the U.S. Department of Energy's Program for Climate Model Diagnosis and Intercomparison provides coordinating support and led development of software infrastructure in partnership with the Global Organization for Earth System Science Portals. ECMWF (Reading, United Kingdom) is acknowledged for access to its ERA-Interim dataset. This work was funded by the Austrian Science Fund (FWF) under

research grants P21642-N21 (TRENDEVAL) and P27724-NBL (VERTICLIM).

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
