# Peer review of "Tropical convection regimes in climate models: evaluation with satellite observations"

_Atmospheric Chemistry and Physics, 2017_

## Referee Comment (RC1) · Anonymous Referee #1 · 18 Oct 2017

**General comments**

Generally I appreciate this work. It is a standing problem to find a method such that climate models can benefit from the accurate GNSS Radio Occultation (RO) technology. Climate models represent the atmosphere in a statistical sense, and they are not expected to correspond to any single measurement at a given time. On the other hand models should be able to resolve processes to some extend in a realistic way, and their ability to do so should be evaluated against state of the art measurements. The method presented in this paper overcomes the gab between models and measurements by extracting classes of profiles belonging to same range of convective activity or "regime" and then collocate RO profiles with model profiles of the same regime. The classification of RO profiles is aided by ERA-Interim analysis. I believe that the method is new

to RO applications, and that it could possibly be applied in studies of other important atmospheric processes than tropical convection and down welling.

The language is generally clear throughout the paper and I did not see any typos. But there is a list of issues about the method which are not sufficiently clearly described in the paper, and which all needs to be addressed before publication can be allowed. These are listed under "Specific comments".

**Specific comments**

p5 l2 (About the specific humidity error estimates): There is a loosely thrown reference to Kursinski and Gebhardt, 2014, citing a systematic error of only 0.03 g/kg for climatological averages. I think that is a bit optimistic, since this estimate is based COSMIC data, while a large part of the RO data used in the analysis comes from the CHAMP mission, known to have sampling issues in the troposphere (also mentioned in Kursinski and Gebhardt, 2014). Why not refer to specific error estimates from the WEGC (OPS 5.6) RO data?

Figure 6 and 7: The error bars show the STD of the "classes" I suppose. I would like to see some indication of the expected RO error, which at least would amount to the mentioned 0.03 g/kg.

Sec. 3.1 line 20 (Collocation in time): RO profiles are matched to the nearest grid point in space and time. I fail to see the purpose of matching the exact time of an event, so what is meant here? Is it a requirement that the RO profile belongs to the same time point in the annual cycle? Please describe clearly what is being done.

The spread in temperature and humidity is surprisingly small in all cases, but especially for the HadGEM2. In figure 4 for example, how can "all classes of $\omega_{500}$ and $T_{2m}$" have a spread of only a couple of degrees? Especially when figure 3 shows that a the spread in $T_{2m}$ is around 10 degrees. Clarification is needed.

Especially the following seems like a paradox: In figure 6 the RO specific humidity is

plotted for all classes. The thickness of the orange area in the 5 plots must reflect the moderate spread of RO specific humidity among the classes. Why is the orange line so narrow in the HadGEM2 plot compared to the rest of the plots? It is based on the same class definitions and the same set of RO profiles after all.

All the climate models, except for HadGEM2, show cold biased tropopause temperature. HadGEM2 sticks out from the other models in many ways besides being least biased in humidity. Some reflection about what could be the reason for that would be appropriate.

This last question is a bit unclear and can maybe not be answered, but please consider it: Since ERA-I resolves updraft / downdraft better than climate models, the classification of RO tends to fall in more extreme vertical winds classes, which are not well represented in the coarse grained climate model. As such a particular vertical wind regime does not necessarily represent the same physical state in ERA-I as it would in a climate model. In that view it may be natural and sound that (some) climate models do not capture the really extreme dry events and therefore appear positively biased in non convective regions.

**Technical comments**

In figure 3, the upper row: It is not really RO data that is used for this plot. It is ERA-I evaluated at RO locations. I suggest to write ERA-I in the figure legend.

Figure 6 and 7: I do not see the purpose of plotting the specific humidity on a reversed (logarithmic) axis?

The curves in Figures 1 are a bit hard to distinguish in a printed version.

I appreciate the color coding of the climate models. Nevertheless the colored font is a bit hard to read in figures 4-7.

Why not compare RO to ERA-I also, just for reference?

---

## Referee Comment (RC2) · Anonymous Referee #2 · 19 Oct 2017

**1  General Comments**

This article demonstrates the use of radio occultation (RO) satellite data for the verification of climate models. In particular, radio occultation from a combination of satellites is used to evaluate tropical temperature and humidity profiles for five atmosphere-only climate simulations.

The research described in this article is original and technically sound, making it suitable for publication in Atmospheric Chemistry and Physics. The quality of the writing is adequate and I think can be improved upon. I did find some sentences difficult to understand, some of which I've highlighted below. However, the article is well organised.

**2  Specific Comments**

1. I don't understand why it is necessary to collocate the RO observations and the climate models. Are there gaps in the spatial/temporal coverage of the RO observations that make this necessary? As co-location requires daily model output, this will make using RO observations a challenge simply in terms of data volumes. How different would the results be without co-location?

2. Given that ERA-Interim is quite widely used to evaluate temperature and humidity in climate mode, it would be interesting (and presumably fairly straightforward) to compare the ERA-Interim temperature and humidity profiles to the OR observations.

3. You mention that a different reanalysis "might give a slightly different distribution of observed RO profiles". Could you try actually redoing this analysis using a different reanalysis?

**3  Technical Corrections**

P1 L18: "and only partly represent high updraft or downdraft velocities". I only understood what you meant by this once I'd read the paper once. Consider rewriting.

P2 L14: "Its proper representation..." It's not clear what it is here.

P2 L18: "Bony et al. (2015)..." This sentence doesn't make sense.

P3 L2: Change "has shown" to "have shown".

P4, L1: You say (paraphrasing) that the quality of RO measurements is best in the upper troposphere and lower stratosphere, but that the uncertainty of individual profiles
* * *
Interactive
comment

is about 0.7 K in the tropopause region and decreases towards the low troposphere. This seems contradictory to me.

P5, L4: I think you mean "Extensive" rather than "Excessive" here.

Table 1: I think there are some inconsistencies between the resolution and the number of latitude/longitude points. E.g. BCC-CSM1 with a longitudinal resolution of 1.875 degrees has 192 longitude points, not 128.

P17, L32: I would change "Model profiles are clustered narrower from..." to "Model profiles are clustered over a narrower pressure range from...".

---

## Author Comment (AC1) · 29 Jan 2018

**Response to Referee#1**

We thank the reviewer for the constructive and thorough review and for very valuable comments.

Please find our detailed response below the reviewer's original comments.

**General comments**

Generally I appreciate this work. It is a standing problem to find a method such that climate models can benefit from the accurate GNSS Radio Occultation (RO) technology. Climate models represent the atmosphere in a statistical sense, and they are not expected to correspond to any single measurement at a given time. On the other hand models should be able to resolve processes to some extend in a realistic way, and their ability to do so should be evaluated against state of the art measurements. The method presented in this paper overcomes the gab between models and measurements by extracting classes of profiles belonging to same range of convective activity or "regime" and then collocate RO profiles with model profiles of the same regime. The classification of RO profiles is aided by ERA-Interim analysis. I believe that the method is new to RO applications, and that it could possibly be applied in studies of other important atmospheric processes than tropical convection and down welling.

The language is generally clear throughout the paper and I did not see any typos. But there is a list of issues about the method which are not sufficiently clearly described in the paper, and which all needs to be addressed before publication can be allowed. These are listed under "Specific comments".

**Specific comments**

**Comment 1 – p5 l2** (About the specific humidity error estimates): There is a loosely thrown reference to Kursinski and Gebhardt, 2014, citing a systematic error of only 0.03 g/kg for climatological averages. I think that is a bit optimistic, since this estimate is based COSMIC data, while a large part of the RO data used in the analysis comes from the CHAMP mission, known to have sampling issues in the troposphere (also mentioned in Kursinski and Gebhardt, 2014). Why not refer to specific error estimates from the WEGC (OPS 5.6) RO data?

**Response 1:**
In our study we do not use RO climatological fields but we use individual RO profiles which we collocate with model profiles. We collect the profiles in classes and compute difference statistics. Thus, the sampling aspect, which is important for computing climatological fields and the related error of climatological averages, is not relevant in this study as we use individual profile collocations.
We thus decided to remove the sentence on the accuracy estimate for climatological averages in the revised manuscript as it might lead to confusion.
We added a reference to a recent publication on tropospheric water vapor (Pincus et al., 2017) which includes a comparison of WEGC OPSv5.6 RO profiles to radiosonde profiles from the Global Reference Upper Air Network (GRUAN). The result shows consistency in specific humidity between the two data sets, which is also in agreement with accuracy estimates for humidity profiles given by Kursinski and

Gebhardt (2014). We also refer to the meanwhile published work of Rieckh et al. (2017) who rigorously compared RO humidity in dry atmospheric layers with observations and reanalyses.

**Page 5, line 2 -4:** We updated the statement in the revised manuscript. It reads now:
"Comparison of WEGC OPS v5.6 RO profiles with collocated radiosonde profiles from the Global Reference Upper Air Network (GRUAN) show consistency within 0.1 g kg$^{-1}$ to 0.3 g kg$^{-1}$ in specific humidity in the middle to lower tropical troposphere (Pincus et al., 2017), confirming accuracy estimates given by Kursinski and Gebhardt (2014). RO data are also useful in the upper troposphere where specific humidity is small. Rieckh et al. (2017) demonstrated that the detection of extremely dry atmospheric layers is possible with RO and showed agreement with data from multiple sources including, e.g., aircraft campaigns and reanalyses."

**Comment 2 – Figure 6 and 7:** The error bars show the STD of the "classes" I suppose. I would like to see some indication of the expected RO error, which at least would amount to the mentioned 0.03 g/kg

**Response 2:**

In Figure 6 and 7 we show the mean profiles of each class and the mean difference profiles of each class which are based on different numbers of collocated profiles in each class (as shown in Figure 3.). The mean difference and the standard deviation of the differences are indicated in the figures. The differences are given in percent.

We considered and discussed the suggestion to indicate an RO error. As discussed above, a climatological error includes the sampling error and is defined for a field of gridded or zonal means. In our case, the visual indication of the RO error would not be appropriate for various reasons:

(1) The climatological error for RO is based on gridded fields and not appropriate for classes of collocated profiles as used in this study.
(2) The numbers of collocations differ from class to class; thus, indicating a statistical error for each mean profile is hardly possible.
(3) The difference profiles are given in percent. However, the statistical error would be quite small due to dividing by the square root of the number of profiles.

**In the captions of Figure 5, 6, and 7** of the revised manuscript we added the following sentence and explicitly state that:
"Mean and standard deviation of the differences are indicated in black."

**Comment 3**

**Comment 3a) Sec. 3.1 line 20 (Collocation in time):** RO profiles are matched to the nearest grid point in space and time. I fail to see the purpose of matching the exact time of an event, so what is meant here? Is it a requirement that the RO profile belongs to the same time point in the annual cycle? Please describe clearly what is being done.

**Response 3a:**

We want to compare the same conditions in observations and atmospheric models, which have sea surface temperature prescribed. Thus, it is important to match the location and time of the day of updraft and downdraft occurrences in order to compare for the same conditions. The collocated grid point in space and time with respect to an RO event was calculated as the minimum

distance between latitude, longitude, and time of the RO observation and the respective latitude, longitude, and time of the model. The time of the RO event is collocated within a time window of plus/minus three hours to model time according to the 6-hourly model resolution. To ensure a correct collocation in daily time, the different calendars of the models need to be taken into account.

**At the beginning of section 3.1** we make this clearer now and included the following explanation:

"It is important to match the location and time of the day of updraft and downdraft occurrences in order to compare for the same conditions in observations and atmospheric models with prescribed sea surface temperature. All available RO profiles for the defined time period and region were selected. The time and location information of each individual RO profile, i.e., longitude and latitude of the tangent point and time of the RO event, was stored for collocation with model data. The collocation procedure was based on a nearest neighbor approach. The collocated grid point in space and time with respect to an RO event was calculated as the minimum distance between latitude, longitude, and time of the RO observation and the respective latitude, longitude, and time of the model. Colocations in time were assigned within a time window of plus/minus three hours according to the 6-hourly model resolution. To ensure a correct collocation in time, the different calendars of the models need to be taken into account as some models use 360 days per year (BCC-CSM1.1, CCSM4, NorESM1-M), 365 days per year (HadGEM2-A), or a standard calendar (GFDL-CM3). Collocated temperature and specific humidity profiles were extracted from RO and from the models at the specific locations. In addition, ERA-Interim 6-hourly data of surface temperature and vertical velocity at 500 hPa were collocated and sampled at the RO event locations. The reanalysis data were only used as proxy for classification of the observations because these variables are not provided by RO."

**Comment 3b:**

The spread in temperature and humidity is surprisingly small in all cases, but especially for the HadGEM2. In figure 4 for example, how can "all classes of omega500 and T2m" have a spread of only a couple of degrees? Especially when figure 3 shows that a the spread in T2m is around 10 degrees. Clarification is needed.

**Response 3b:**

In Fig. 3 we show the number of data falling into the different classes for observations and models. The classes are defined for surface temperatures and range from 18°C to 32°C.
In all following figures we compare the *mean* profiles per class *above* the boundary layer. RO information below the boundary layer is limited as most profiles do not reach to the surface. We thus do not use the RO profiles below 2 km or below about 800 hPa in our comparison. Near this height/pressure levels in the free atmosphere, the spread in mean atmospheric temperature profiles becomes smaller (about half) compared to the spread of the surface temperatures.
To make this clear for the reviewer, we show in Fig. 1 to Fig. 4 below, individual profiles sampled into the different surface temperature classes and the class mean temperatures. The plots are shown for a vertical velocity class of 30 hPa where the spread in temperature classes and the occupation with data is largest. We step through each of the 15 surface temperature classes. Figure A1 shows the RO profiles as function of altitude from the surface to 5 km. Figure A2 shows the corresponding profiles of the HadGEM2 model. Figure A3 shows the RO profiles as function of pressure up to 500 hPa. Figure A4 shows the corresponding profiles of NorESM as function of pressure. The spread in temperature is clearly broader near the surface than in the free atmosphere. Furthermore, it is shown that the mean profiles of each class match the surface classes.

**Comment 3c:**

Especially the following seems like a paradox: In figure 6 the RO specific humidity is plotted for all classes. The thickness of the orange area in the 5 plots must reflect the moderate spread of RO specific humidity among the classes. Why is the orange line so narrow in the HadGEM2 plot compared to the rest of the plots? It is based on the same class definitions and the same set of RO profiles after all.

**Response 3c:**

We thank the reviewer for pointing to this issue. Re-checking the code, we found that this is actually due to a different subsample used for HadGEM due to a technical issue. The HadGEM2 data files are of larger size because they contain more vertical levels than the other models. Thus, we split them and handled them differently in parts of the computations, which led to this error.

We corrected this. We re-computed everything and replotted all figures based on the full sample size of HadGEM2. This is reflected in a larger spread of the profiles comparable to the other models. The resulting temperature and specific humidity differences show a larger standard deviation while the characteristics in the mean difference to RO did not change in Fig. 5 and Fig. 7. Main changes are seen in Fig. 7. The differences to RO are larger and especially for downdraft over sea the behavior is similar to that of the other models which is reasonable. We corrected the figures and the corresponding text in the revised manuscript accordingly. However, this did not change the main findings of this study.

**Fig.4 to Fig. 7:** We corrected the figures with corrected figures for HadGEM based on the full sample size.

**Section 4.2, paragraph 3: "**Over land, the agreement is best with mean differences of less than 10%, except for GFDL_CM3. In convection regions over the ocean, most models tend to underestimate moisture by 10% to 30%. HadGEM2-A shows the best performance in representation of specific humidity…"

We corrected the corresponding text to:

"Over land, the agreement is best with mean differences of 10% to 20%, except for GFDL_CM3. In convection regions over the ocean, most models tend to underestimate moisture by 10% to 40%. NorESM1-M shows the best performance in representation of specific humidity…"

**Section 4.2, last paragraph:** "Largest differences are found in non-convection regions over the ocean where all models, except HadGEM2-A, …"

We corrected the text. It now reads:

"Largest differences are found in non-convection regions over the ocean where all models …"

**Section 5, paragraph 6: "**Specific humidity differences of about only 10% are found for HadGEM2 and NorESM, whereas BCC-CSM1.1 and CCSM4 are about 20% to 30% too dry over sea during convection…"

We corrected the corresponding text to:

**"**Specific humidity differences of about only 10% are found for NorESM1-M, whereas BCC-CSM1.1, CCSM4, and HadGEM2 are about 20% to 40% too dry over sea during convection…"

**Comment 3d:**

All the climate models, except for HadGEM2, show cold biased tropopause temperature. HadGEM2 sticks out from the other models in many ways besides being least biased in humidity. Some reflection about what could be the reason for that would be appropriate.

**Response 3d:**

Specific humidity in HadGEM2 shows a similar behavior as the other models after correcting for the full sample size (please see response 3c above.)

The HadGEM2 model shows actually a warm bias near the tropopause. We state this in section 4.1, paragraph 2, and also discuss that the bias is reduced in the new model version (HadGEM3), which has been published meanwhile by Williams et al. (2018). They state that the warm bias "…has been considerably reduced, primarily through a change to the stencil used by the semi-lagrangian dynamics in the calculation of the departure point for the vertical advection of potential temperature to make it more accurate in regions of large gradients…" (see Williams et al., 2018, their Fig. 8).

**Section 4.1, paragraph 2:** We included a citation to Williams et al. (2018).

**Comment 3e:**

This last question is a bit unclear and can maybe not be answered, but please consider it: Since ERA-I resolves updraft / downdraft better than climate models, the classification of RO tends to fall in more extreme vertical winds classes, which are not well represented in the coarse grained climate model. As such a particular vertical wind regime does not necessarily represent the same physical state in ERA-I as it would in a climate model. In that view it may be natural and sound that (some) climate models do not capture the really extreme dry events and therefore appear positively biased in non convective regions.

**Response 3e:**

We thank the reviewer for this interesting comment. We agree that some bias might be expected because of this effect as the reviewer suggests. We included a respective statement in section 5 of the revised manuscript.

**Section 5, paragraph 7:** "Largest differences are found in non-convection regions over sea where almost all models are too wet and overestimate observed moisture by 100%. This appears reasonable as some climate models do not capture the really extreme dry events and therefore appear positively biased in non-convective regions."

**Technical comments**

**Comment – In figure 3, the upper row:** It is not really RO data that is used for this plot. It is ERA-I evaluated at RO locations. I suggest to write ERA-I in the figure legend.

**Response:** In Fig. 3, top row, we show the number of RO profiles that are sampled into updraft and downdraft classes. Only the classes are defined by using sea surface temperature and vertical velocity from ERA-Interim. The figure legend is correct.

**Comment – Figure 6 and 7:** I do not see the purpose of plotting the specific humidity on a reversed (logarithmic) axis?

**Response:** We agree that it is counterintuitive but the purpose was to have it least over plotted by the difference profiles. We now changed the figures according to your suggestion.

**Fig. 6 and Fig. 7:** We replotted Fig. 6 and Fig.7 and reversed the axis for specific humidity.

**Comment – The curves in Figures 1** are a bit hard to distinguish in a printed version. I appreciate the color coding of the climate models. Nevertheless the colored font is a bit hard to read in figures 4-7.

**Response:** We assume that Fig. 2 is meant here. The main purpose of this figure is to show basic similarities and differences in the distributions of surface temperature and vertical velocity which are used for classification. We think the figure serves this purpose.

**Fig. 2 to Fig. 7:** We improved the resolution of the figures.

**Comment:** Why not compare RO to ERA-I also, just for reference?

**Response:** We agree that this might be interesting. However, there are several studies which compare temperature and humidity from RO observations with analyses or reanalyses and which we cite in our manuscript, e.g., Rieckh et al. (2017) use ERA-Interim, Vergados et al. (2016) use MERRA, and Pincus et al. (2017) provide an overview on the representation of tropospheric water vapor in analyses and reanalyses.

We find a comparison to reanalyses beyond the scope of this study. In the current study setup, 6-hourly gridded fields and high vertical resolution are required. For a reasonable comparison with ERA-Interim one would choose a vertical gridding comparable with RO. The file sizes including a few years of data gets very large, reaching more than 80 GB, which is hard to handle in the current setup of the study based on using individual profiles and would require a redesign.

For the reviewers convenience we show in Fig. A5 below temperature differences of ERA-Interim minus RO, which are based on monthly-mean zonal-mean climatological fields with a vertical gridding of 200 m. The distinct differences in the tropopause region and in the stratosphere stems from a known bias of ERA-Interim (Poli et al. 2010; S. Healy (ECMWF) pers. comm.).

**References:**

Pincus, R., Beljaars, A., Buehler, S. A., Kirchengast, G., Ladstaedter, F, and Whitaker, J. S.: The Representation of Tropospheric Water Vapor Over Low-Latitude Oceans in (Re-)analysis: Errors, Impacts, and the Ability to Exploit Current and Prospective Observations, Surv. Geophys., 38, 1399–1423, doi:10.1007/s10712-017-9437-z, 2017.

Poli, P., Healy, S. B., and Dee, D. P.: Assimilation of Global Positioning System radio occultation data in the ECMWF ERA–Interim reanalysis, Q. J. R. Meteorol. Soc., 136, 1972–1990, doi:10.1002/qj.722, 2010.

Rieckh, T., Anthes, R., Randel, W., Ho, S.-P. and Foelsche, U.: Tropospheric dry layers in the tropical western Pacific: comparisons of GPS radio occultation with multiple data sets, Atmos. Meas. Tech., 10, 1093–1110, doi:10.5194/amt-10-1093-2017, 2017.

Williams, K. D., et al.: The Met Office Global Coupled Model 3.0 and 3.1 (GC3.0 & GC3.1) Configurations, J. Adv. Model. Earth Syst., doi:10.1002/2017MS001115, 2018.

[Figure]

**Fig.A1** RO profiles as function of altitude (orange) and mean profiles (black) sampled into classes. Shown for 15 surface temperature classes (x) and for a vertical velocity class of 30 hPa.

[Figure]

**Fig.A2** HadGEM2 profiles as function of altitude (green) and mean profiles (black) sampled into classes. Shown for 15 surface temperature classes (x) and for the vertical velocity class of 30 hPa.

[Figure]

**Fig.A3** Same as Fig.1 but RO profiles as function of pressure (orange) and mean profiles (black).

[Figure]

**Fig.4** NorESM profiles as function of pressure (magenta) and mean profiles (black) sampled into classes. Shown for 15 surface temperature classes (x) and for the vertical velocity class of 30 hPa.

[Figure]

**Fig.A5** Monthly mean temperature differences or ERA-Interim minus RO.

---

## Author Comment (AC2) · 29 Jan 2018

**Response to Referee#2**

We thank the reviewer for the constructive review, the interesting questions, and very valuable comments.

Please find our detailed response below the reviewer's original comments.

**1 General Comments:**

This article demonstrates the use of radio occultation (RO) satellite data for the verification of climate models. In particular, radio occultation from a combination of satellites is used to evaluate tropical temperature and humidity profiles for five atmosphere-only climate simulations.

The research described in this article is original and technically sound, making it suitable for publication in Atmospheric Chemistry and Physics. The quality of the writing is adequate and I think can be improved upon. I did find some sentences difficult to understand, some of which I've highlighted below. However, the article is well organised.

**2 Specific Comments:**

**Comment 1:** I don't understand why it is necessary to collocate the RO observations and the climate models. Are there gaps in the spatial/temporal coverage of the RO observations that make this necessary? As co-location requires daily model output, this will make using RO observations a challenge simply in terms of data volumes. How different would the results be without co-location?

**Response 1:**

Radio occultation observations are based on a limb sounding technique, which provides discrete measurements in form of vertical profiles. The observations are globally well distributed. For the reviewer's convenience we show the global coverage with RO measurements for one exemplary day in Fig. A1 below. One of the strengths of RO is the good vertical resolution.

In this study we use individual RO profiles and compare them to atmospheric models at the best available vertical resolution. We want to compare the same conditions in observations and atmospheric models, which have sea surface temperature prescribed. Thus, it is important to match the location and time of the day of updraft and downdraft occurrences in order to compare for the same conditions.

This requires, as the reviewer noted, a daily/6-hourly model output and data files get quite large. Thus, the comparison is computationally intensive in our current study setup.

However, besides individual profiles, RO data can also be sampled and averaged to climatological fields. Thus, another approach for model comparison could be gridded RO fields, which are regularly used in form of monthly-mean zonal-mean fields available for the whole RO record. From 2006 onwards, the data amount is large enough for a finer gridding because data from more than a single RO satellite are available. Daily climatologies of 2.5° x 2.5° in latitude and longitude are possible but require a weighted

averaging in space and time (weighting over +/-1 day). The use of such higher resolved daily RO climatological fields has been demonstrated for the investigation of atmospheric blocking recently (see Brunner et al., 2016; Brunner and Steiner, 2017).

As these higher resolved RO climatological fields are available only from 2006 onward (and only since recently) and the AMIP data are available only until 2008, we decided to use individual profiles for model comparison in this study. Furthermore, a daily to sub-daily resolution is a prerequisite to resolve and compare for different atmospheric conditions, as done in this study.

**In section 5, we added the following paragraph:**

"RO data are available as individual profiles and gridded climatological fields. The latter are commonly available in form of monthly-mean zonal-mean climatologies for the whole RO record. From 2006 onward, the data amount is large enough for a finer gridding because data from more than a single RO satellite are available. Daily climatologies of 2.5° x 2.5° in latitude and longitude are possible but require a weighted averaging in space and time. The use of such higher resolved daily RO climatological fields has been demonstrated only recently for the investigation of atmospheric blocking (see Brunner et al., 2016; Brunner and Steiner, 2017)."

**Comment 2:** Given that ERA-Interim is quite widely used to evaluate temperature and humidity in climate mode, it would be interesting (and presumably fairly straightforward) to compare the ERA-Interim temperature and humidity profiles to the OR observations.

You mention that a different reanalysis "might give a slightly different distribution of observed RO profiles". Could you try actually redoing this analysis using a different reanalysis?

**Response 2:**

We agree that this might be interesting. However, there are already several studies which compare temperature and humidity from RO observations with analyses or reanalyses and which we cite in our manuscript, e.g., Rieckh et al. (2017) use ERA-Interim, Vergados et al. (2016) use MERRA, and Pincus et al. (2017) provide an overview on the representation of tropospheric water vapor in analyses and reanalyses.

We find a comparison to reanalyses beyond the scope of this study. In the current study setup, 6-hourly gridded fields and high vertical resolution are required. For a reasonable comparison with ERA-Interim one would choose a vertical gridding comparable with RO. The size of files including a few years of data gets very large, reaching more than 80 GB, which is hard to handle in the current setup of the study and would require a redesign.

For the reviewers convenience we show in Fig.A2 below temperature differences of ERA-Interim minus RO which are based on monthly-mean zonal-mean climatological fields and a vertical gridding of 200 m. There are distinct differences in the tropopause region and in the stratosphere, which stem from a known bias of ERA-Interim (Poli et al. 2010; S. Healy (ECMWF) pers. comm.).

**2 Technical Corrections:**

**Comment P1 L18:** "and only partly represent high updraft or downdraft velocities". I only under-stood what you meant by this once I'd read the paper once. Consider rewriting.

**Response P1 L18:** We rephrased the sentence to make it clearer. The sentence now reads: "and only partly represent strong vertical wind classes."

**Comment P2 L14:** "Its proper representation. . ." It's not clear what it is here.

**Response P2 L14:** We rewrote the sentence to make it clearer. It reads now: "The proper representation of the tropospheric structure in climate models is of central importance since it has more impact than other regions of the atmosphere.

**Comment P2 L18:** "Bony et al. (2015). . ." This sentence doesn't make sense.

**Response P2 L18:** We changed the sentence to: "Bony et al. (2015) point in particular to enhance the understanding of cloud feedbacks and convective organization."

**Comment P3 L2:** Change "has shown" to "have shown".

**Response P3 L2:** We changed it and write "have shown".

**Comment P4, L1:** You say (paraphrasing) that the quality of RO measurements is best in the upper troposphere and lower stratosphere, but that the uncertainty of individual profiles is about 0.7 K in the tropopause region and decreases towards the low troposphere. This seems contradictory to me.

**Response :**

In section 2.1 of the manuscript we explain the retrieval of dry and physical atmospheric RO parameters in some detail. In a dry atmosphere, where water vapor is negligible, RO dry temperature profiles can be retrieved without further background information. In the troposphere, temperature and humidity are retrieved based on optimal estimation of RO and background profiles.

For RO profiles the uncertainty is lowest in the upper troposphere and lower stratosphere region. Above this region the observational error increases into the stratosphere. For dry temperature profiles, the observational error increases also in the lower troposphere (see Scherllin-Pirscher et al., 2011a). For physical temperature profiles (mainly used in this study), the observational error slightly decreases in the troposphere. The reason for decreasing RO temperature errors in the lower troposphere is the increasing influence of background information in the blended product. RO error estimates for physical temperature are given by Scherllin-Pirscher et al. (2017) for the Wegener Center RO data.

**P4, L1:** We removed "and slightly decreases toward the lower troposphere". We agree that it is confusing. We refer to the work of Scherllin-Pirscher et al. (2017) instead. The sentence reads now:

"The observational uncertainty of individual temperature profiles is about 0.7 K in the tropopause region and detailed estimates are given by Scherllin-Pirscher et al. (2017)."

**Comment P5, L4:** I think you mean "Extensive" rather than "Excessive" here.

**Response P5, L4::** We corrected the sentence and write "Extensive".

**Comment Table 1:** I think there are some inconsistencies between the resolution and the number of latitude/longitude points. E.g. BCC-CSM1 with a longitudinal resolution of 1.875 degrees has 192 longitude points, not 128.

**Response Table 1:** Thank you for pointing to this error. The resolution of BCC-CSM1 is 2.8125 degrees in longitude and 2.8125 in latitude, which corresponds to 128 x 64 points. We thoroughly checked all other numbers in Table 1 and found them correct.

We made the following corrections in the manuscript:
**Table 1, line BCC-CSM1.1, column 2:** We corrected the horizontal resolution to 2.8125° x 2.8125° (126 x 64).

**P5, L25:** We changed the sentence on the models' horizontal resolution ranges accordingly: "The models' horizontal resolution ranges from near 1.25° x 0.95° to 2.8° x 2.8° in longitude and latitude."

**Comment P17, L32:** I would change "Model profiles are clustered narrower from. . ." to "Model profiles are clustered over a narrower pressure range from. . ."

**Response P17, L32:** We changed the sentence as suggested.

**References:**

Brunner, L. and Steiner, A. K.: A global perspective on atmospheric blocking using GPS radio occultation – one decade of observations, Atmos. Meas. Techn., 10, 4727–4745, doi:10.5194/amt-10-4727-2017, 2017.

Brunner, L., Hegerl, G. C. and Steiner, A. K.: Connecting atmospheric blocking to European temperature extremes in spring, J. Climate, 30, 585–594, doi: 10.1175/JCLI-D-16-05181, 2017.

Pincus, R., Beljaars, A., Buehler, S. A., Kirchengast, G., Ladstaedter, F, and Whitaker, J. S.: The Representation of Tropospheric Water Vapor Over Low-Latitude Oceans in (Re-)analysis: Errors, Impacts, and the Ability to Exploit Current and Prospective Observations, Surv. Geophys., 38, 1399–1423, doi:10.1007/s10712-017-9437-z, 2017.

Poli, P., Healy, S. B., and Dee, D. P.: Assimilation of Global Positioning System radio occultation data in the ECMWF ERA–Interim reanalysis, Q. J. R. Meteorol. Soc., 136, 1972–1990, doi:10.1002/qj.722, 2010.

Rieckh, T., Anthes, R., Randel, W., Ho, S.-P. and Foelsche, U.: Tropospheric dry layers in the tropical western Pacific: comparisons of GPS radio occultation with multiple data sets, Atmos. Meas. Tech., 10, 1093–1110, doi:10.5194/amt-10-1093-2017, 2017.

Scherllin-Pirscher, B., Steiner, A. K., Kirchengast, G., Kuo, Y.-H. and Foelsche, U.: Empirical analysis and modeling of errors of atmospheric profiles from GPS radio occultation, Atmos. Meas. Tech., 4(9), 1875–1890, doi:10.5194/amt-4-1875-2011, 2011a.

Scherllin-Pirscher, B., Steiner, A. K., Kirchengast, G., Schwärz, M. and Leroy, S. S.: The power of vertical geolocation of atmospheric profiles from GNSS radio occultation, J. Geophys. Res. Atmos., 2016JD025902, doi:10.1002/2016JD025902, 2017.

[Figure]

**Fig. A1** Global distribution of radio occultation measurements on January 1, 2010.

[Figure]

**Fig.A2** Monthly mean temperature differences or ERA-Interim minus RO.

---

## Author Response (AR2)

**Manuscript number acp-2017-669 "Tropical convection regimes in climate models: evaluation with satellite observations" *by* Andrea K. Steiner et al.**

**Response to Editor:**

We thank the editor for the thorough handling of our manuscript and for the valuable comments. Please find our detailed response below the editor's original comments.

**Comment 1:** In Figs 6 and 7 the error bars look symmetric around the mean although the values are plotted on a logarithmic scale. Is this by intention? This would mean that the errors on the left (negative) side are much smaller than those on the right (positive side). This is not consistent with the usual definition of the standard deviation. Could it be that the error bars you show are the standard deviation of log(delta_specific_humidity)? If so you need to clearly note that.

**Response 1:**
In Figs 6 and 7, the specific humidity differences (and standard deviation) are plotted in percent and refer to the upper x-axis (which is not logarithmic). Only the collocated specific humidity profiles of RO and models are plotted on a logarithmic scale and refer to the lower (logarithmic) x-axis.

**Captions of Fig. 6 and 7:** For better clarity we explicitly point to the respective x-axis in the revised manuscript text:
"Figure 6. Specific humidity difference of model data minus RO observations (upper axes) and collocated specific humidity profiles (lower axes) for all classes ($\omega_{500}$ and $T_{2m}$) in the tropics, ..."
"Figure 7. Specific humidity difference of model data minus RO observations (upper axes) and collocated specific humidity profiles (lower axes) for BCC-CSM1.1, CCSM4, GFDL-CM3, NorESM1-M, and HadGEM2 (top to bottom),..."
**Fig. 7:** We corrected the units in the lower x-axis title to "(g/kg)" (it was erroneously printed as "(%)" in the former version of this figure).

**Comment 2:** You mention that for finding collocations between the RO measurements and profiles from the model calculations, you calculate the minimum distance between latitude, longitude, and time of the RO observation and the respective latitude, longitude, and time of the model. How do you calculate this norm exactly, given that distance and time difference have different units? Please explain this in the text.

**Response 2:**
We thank you for this comment and agree that the formulation "minimum distance" is not clear.

[revised manuscript text omitted]